# CATER: A DIAGNOSTIC DATASET FOR COMPOSITIONAL ACTIONS & TEMPORAL REASONING

**Rohit Girdhar**[1][*]    **Deva Ramanan**[1,2]
[1]Carnegie Mellon University    [2]Argo AI
http://rohitgirdhar.github.io/CATER

## ABSTRACT

Computer vision has undergone a dramatic revolution in performance, driven in large part through deep features trained on large-scale supervised datasets. However, much of these improvements have focused on static image analysis; video understanding has seen rather modest improvements. Even though new datasets and spatiotemporal models have been proposed, simple frame-by-frame classification methods often still remain competitive. We posit that current video datasets are plagued with implicit biases over scene and object structure that can dwarf variations in temporal structure. In this work, we build a video dataset with fully observable and controllable object and scene bias, and which truly requires spatiotemporal understanding in order to be solved. Our dataset, named CATER, is rendered synthetically using a library of standard 3D objects, and tests the ability to recognize compositions of object movements that require long-term reasoning. In addition to being a challenging dataset, CATER also provides a plethora of diagnostic tools to analyze modern spatiotemporal video architectures by being completely observable and controllable. Using CATER, we provide insights into some of the most recent state of the art deep video architectures.

## 1 INTRODUCTION

While deep features have revolutionized static image analysis, video descriptors have struggled to outperform classic hand-crafted descriptors (Wang & Schmid, 2013). Though recent works have shown improvements by merging image and video models by inflating 2D models to 3D (Carreira & Zisserman, 2017; Feichtenhofer et al., 2016), simpler 2D models (Wang et al., 2016b) still routinely appear among top performers in video benchmarks such as the Kinetics Challenge at CVPR'17. This raises the natural question: are videos trivially understandable by simply averaging the predictions over a sampled set of frames?

At some level, the answer must be no. Reasoning about high-level cognitive concepts such as intentions, goals, and causal relations requires reasoning over long-term temporal structure and order (Shoham, 1987; Bobick, 1997). Consider, for example, the movie clip in Fig. 1 (a), where an actor leaves the table, grabs a firearm from another room, and returns. Even though no gun is visible in the final frames, an observer can easily infer that the actor is surreptitiously carrying the gun. Needless to say, any single frame from the video seems incapable of supporting that inference, and one needs to reason over space and time in order to reach that conclusion.

As a simpler instance of the problem, consider the cup-and-balls magic routine[1], or the gambling-based shell game[2], as shown in Fig. 1 (b). In these games, an operator puts a target object (ball) under one of multiple container objects (cups), and moves them about, possibly revealing the target at various times and recursively containing cups within other cups. The task at the end is to tell which of the cups is covering the ball. Even in its simplest instantiation, one can expect any human or computer system that solves this task to require the ability to model state of the world over long temporal horizons, reason about occlusion, understand the spatiotemporal implications of containment, etc. An important aspect of both our motivating examples is the adversarial nature of the task,

---

[*]Now at Facebook AI Research

[1]https://en.wikipedia.org/wiki/Cups_and_balls
[2]https://en.wikipedia.org/wiki/Shell_game

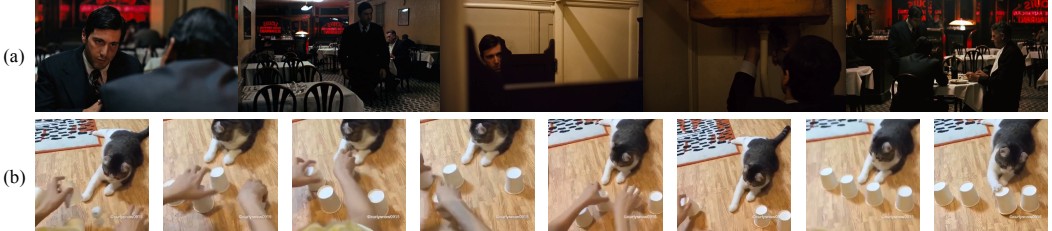

Figure 1: **Real world video understanding.** Consider this iconic movie scene from The Godfather in (a), where the protagonist leaves the table, goes to the bathroom to extract a hidden firearm, and returns to the table presumably with the intentions of shooting a person. While the gun itself is visible in only a few frames of the whole clip, it is trivial for us to realize that the protagonist has it in the last frame. An even simpler instantiation of such a reasoning task could be the cup-and-ball shell game in (b), where the task is to determine which of the cups contain the ball at the end of the trick. **Can we design similarly hard tasks for computers?**

where the operator in control is trying to make the observer fail. Needless to say, a frame by frame prediction model would be incapable of solving such tasks.

Given these motivating examples, why don't spatiotemporal models dramatically outperform their static counterparts for video understanding? We posit that this is due to limitations of existing video benchmarks. Even though video datasets have evolved from the small regime with tens of labels (Soomro et al., 2012; Kuehne et al., 2011; Schuldt et al., 2004) to large with hundreds of labels (Sigurdsson et al., 2016; Kay et al., 2017), tasks have remained highly correlated to the scene and object context. For example, it is trivial to recognize a swimming action given a swimming pool in the background (He et al., 2016b). This is further reinforced by the fact that state of the art pose-based action recognition models (Yan et al., 2018) are outperformed by simpler frame-level models (Wang et al., 2016b) on the Kinetics (Kay et al., 2017) benchmark, with a difference of nearly 45% in accuracy! Sigurdsson *et al.* also found similar results for their Charades (Sigurdsson et al., 2016) benchmark, where adding ground truth object information gave the largest boosts to action recognition performance (Sigurdsson et al., 2017).

In this work, we take an alternate approach to developing a video understanding dataset. Inspired by the recent CLEVR dataset (Johnson et al., 2017) (that explores spatial reasoning in tabletop scenes) and inspired by the adversarial parlor games above (that require temporal reasoning), we introduce **CATER**, a diagnostic dataset for Compositional Actions and TEmporal Reasoning in dynamic tabletop scenes. We define three tasks on the dataset, each with an increasingly higher level of complexity, but set up as classification problems in order to be comparable to existing benchmarks for easy transfer of existing models and approaches. Specifically, we consider primitive action recognition, compositional action recognition, and adversarial target tracking under occlusion and containment. However, note that this does not limit the usability of our dataset to these tasks, and we provide full metadata with the rendered videos that can be used for more complex, structured prediction tasks like detection, tracking, forecasting, and so on. Our dataset does not model an operator (or hand) moving the tabletop objects, though this could be simulated as well in future variants, as in (Rogez et al., 2015).

Being synthetic, CATER can easily be scaled up in size and complexity. It also allows for detailed model diagnostics by controlling various dataset generation parameters. We use CATER to benchmark state-of-the-art video understanding models (Wang et al., 2018; 2016b; Hochreiter & Schmidhuber, 1997), and show even the best models struggle on our dataset. We also uncover some insights into the behavior of these models by changing parameters such as the temporal duration of an occlusion, the degree of camera motion, etc., which are difficult to both tune and label in real-world video data.

## 2 RELATED WORK

**Spatiotemporal networks:** Video understanding for action recognition has evolved from iconic hand-designed models (Wang & Schmid, 2013; Laptev, 2005; Wang et al., 2011) to sophisticated

| Dataset | Size | Len | Task | #cls | TO | STR | LTR | CSB |
|---|---|---|---|---|---|---|---|---|
| UCF101 (Soomro et al., 2012) | 13K | 7s | cls | 101 | ✗ | ✗ | ✗ | ✗ |
| HMDB51 (Kuehne et al., 2011) | 5K | 4s | cls | 51 | ✗ | ✗ | ✗ | ✗ |
| Kinetics (Kay et al., 2017) | 300K | 10s | cls | 400 | ✗ | ✓ | ✗ | ✗ |
| AVA (Gu et al., 2018) | 430 | 15m | det | 80 | ✗ | ✓ | ✗ | ✗ |
| VLOGs (Fouhey et al., 2018) | 114K | 10s | cls | 30 | ✗ | ✓ | ✗ | ✗ |
| DAHLIA (Vaquette et al., 2017) | 51 | 39m | det | 7 | ✓ | ✓ | ✓ | ✗ |
| TACoS (Regneri et al., 2013) | 127 | 6m | align | - | ✓ | ✓ | ✓ | ✗ |
| DiDeMo (Anne Hendricks et al., 2017) | 10K | 30s | align | - | ✓ | ✓ | ✓ | ✗ |
| Charades (Sigurdsson et al., 2016) | 10K | 30s | det | 157 | ✓ | ✓ | ✗ | ✗ |
| Something Something (Goyal et al., 2017) | 108K | 4s | cls | 174 | ✓ | ✓ | ✗ | ✓ |
| Diving48 (Li et al., 2018) | 18K | 5s | cls | 48 | ✓ | ✓ | ✗ | ✓ |
| Cooking (Rohrbach et al., 2012a) | 44 | 3-41m | cls | 218 | ✓ | ✓ | ✗ | ✓ |
| IKEA (Toyer et al., 2017) | 101 | 2-4m | gen | - | ✓ | ✓ | ✓ | ✓ |
| Composite (Rohrbach et al., 2012b) | 212 | 1-23m | cls | 44 | ✓ | ✓ | ✓ | ✓ |
| TFGIF-QA (Jang et al., 2017) | 72K | 3s | qa | - | ✓ | ✓ | ✗ | ✗ |
| MovieQA (Tapaswi et al., 2016) | 400 | 200s | qa | - | ✓ | ✓ | ✗ | ✗ |
| Robot Pushing (Finn et al., 2016) | 57K | 1s | gen | - | ✓ | ✓ | ✗ | ✓ |
| SVQA (Song et al., 2018) | 12K | 4s | qa | - | ✓ | ✓ | ✗ | ✓ |
| Moving MNIST (Srivastava et al., 2015) | - | 2s | gen | - | ✓ | ✓ | ✗ | ✓ |
| Flash MNIST (Long et al., 2018) | 100K | 2s | cls | 1024 | ✗ | ✓ | ✗ | ✓ |
| CATER (ours) | 5.5K | 10s | cls | 36-301 | ✓ | ✓ | ✓ | ✓ |

Table 1: **CATER vs previous datasets** in terms of size (number of videos), average video length, task (**cls**sification, **det**ection, **gen**erative modeling, **align**ment of descriptions, **q**uestion **a**nswering), number of classes; whether tasks require Temporal Ordering (TO), Short Term Reasoning (STR), Long Term Reasoning (LTR); and if the data Controls for Scene Biases (CSB).

spatiotemporal deep networks (Carreira & Zisserman, 2017; Simonyan & Zisserman, 2014; Girdhar et al., 2017; Wang et al., 2018; Xie et al., 2017; Tran et al., 2018; 2015). While similar developments in the image domain have lead to large improvements on tasks like classification (Szegedy et al., 2016; He et al., 2016a; Huang et al., 2017) and localization (He et al., 2017; Papandreou et al., 2017), video models have struggled to out-perform previous hand-crafted descriptors (Wang & Schmid, 2013). Even within the set of deep video architectures, models capable of temporal modeling, such as RNNs (Karpathy et al., 2014) and 3D convolutions (Tran et al., 2015; Varol et al., 2017a) have not shown significantly better performance than much simpler, per-frame prediction models, such as variants of two-stream architectures (Wang et al., 2016b; Simonyan & Zisserman, 2014). Though some recent works have shown improvements by merging image and video models by inflating 2D models to 3D (Carreira & Zisserman, 2017; Feichtenhofer et al., 2016), simple 2D models (Wang et al., 2016b) were still among the top performers in the Kinetics Challenge at CVPR'17.

**Video action understanding datasets:** There has been significant effort put forth to collecting video benchmarks. One line of attack employs human actors to perform scripted actions. This is typically done in controlled environments (Schuldt et al., 2004; Shahroudy et al., 2016; Ionescu et al., 2014), but recent work has pursued online crowd sourcing (Goyal et al., 2017; Sigurdsson et al., 2016). Another direction collects videos from movies and online sharing platforms. Many popular video benchmarks follow this route for diverse, in-the-wild videos, such as UCF-101 (Soomro et al., 2012), HMDB-51 (Kuehne et al., 2011) and more recently Kinetics (Kay et al., 2017) and VLOGs (Fouhey et al., 2018). As discussed earlier, such datasets struggle with the strong bias of actions with scenes and objects. Our underlying thesis is that the field of video understanding is hampered by such biases because they favor image-based baselines. While some recent work (Goyal et al., 2017; Li et al., 2018) attempts to control for this bias, it still remains a challenge for long-term reasoning tasks. One might argue that since such biases are common in the visual world, video benchmarks should reflect them. We take the view that a diverse set of benchmarks are needed to enable comprehensive diagnostics and validation of the state-of-affairs in video understanding. Table 1 shows that CATER fills a missing gap in the benchmark landscape, most notably because of its size/video length, label distribution, relative resilience to object and scene bias, and diagnostic abilities.

**Synthetic data in computer vision:** Our work, being synthetically generated, is also closely related to other works in using synthetic data for computer vision applications. There has been a large body of work in this direction, with the major focus on using synthetic training data for real world applications. This includes semantic scene understanding (Dosovitskiy et al., 2017; Shah et al., 2018; Richter et al., 2017), 3D scene understanding (Girdhar et al., 2016; Su et al., 2015; Wu et al., 2016; Song et al., 2017), human understanding (Varol et al., 2017b; De Souza et al., 2017), optical flow (Butler et al., 2012; Mayer et al., 2016) and navigation, RL or embodied learning (Wu et al., 2018; Kolve et al., 2017; Kempka et al., 2016; Mnih et al., 2013). Our work, on the other hand, attempts to develop a benchmark for video based action understanding. Similar attempts have been made for scene understanding through abstract scenes (Zitnick et al., 2016), with more recently

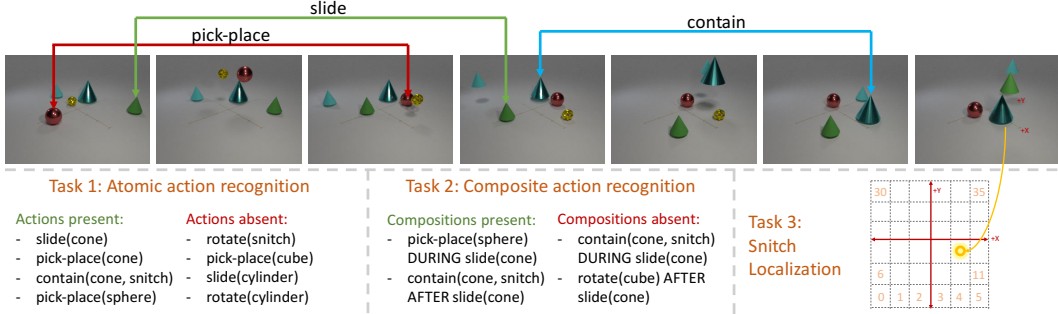

Figure 2: **CATER dataset and tasks.** Sampled frames from a random video from CATER. We show some of the actions afforded by objects in this video, as labeled on the top using arrows. We define three tasks on these videos. Task 1 requires identifying all active actions in the video. Task 2 requires identifying all active compositional actions. Task 3 requires quantized spatial localization of the snitch object at the end of the video. Note that, as in this case, the snitch may be occluded or 'contained' by another object, and hence models would require spatiotemporal understanding to complete the task. Please refer to the **supplementary** video for more example videos.

focusing on building a complex reasoning benchmark, CLEVR (Johnson et al., 2017). In the video domain, benchmarks such as Flash-MNIST (Long et al., 2018), Moving MNIST (Srivastava et al., 2015) and SVQA (Song et al., 2018) have been proposed. Concurrent to us, CLEVRER (Yi et al., 2020), PHYRE (Bakhtin et al., 2019), COPHY (Baradel et al., 2020) and IntPhys (Riochet et al., 2018) benchmarks have been proposed with a focus on causal physical reasoning through QA, RL, prediction and ranking interfaces respectively. On the other hand, CATER focuses on spatiotemporal video reasoning tasks building upon CLEVR, with a simple classification interface, making it easily amenable for existing video understanding systems.

**Object tracking:** Detecting and tracking objects has typically been used as an initial representation for long-term video and activity understanding (Shet et al., 2005; Hongeng et al., 2004; Lavee et al., 2009). Extensions include adversarial tracking, where the objects are designed to be hidden from plain view. It has typically been used for tasks such as determining if humans are carrying an object (Dondera et al., 2013; Ferrando et al., 2006) or abandoned / exchanging objects (Tian et al., 2011; Li et al., 2006). We embrace this direction of work and include state-of-the-art deep trackers (Zhu et al., 2018) in our benchmark evaluation.

## 3 THE CATER DATASET

CATER provides a video understanding dataset that requires long term temporal reasoning to be solved. Additionally, it provides diagnostic tools that can evaluate video models in specific scenarios, such as with or without camera motion, with varying number of objects and so on. This control over the dataset parameters is achieved by synthetically rendering the data. These videos come with a ground truth structure that can be used to design various different video understanding tasks, including but not limited to object localization and spatiotemporal action composition. Unlike existing video understanding benchmarks, this dataset is free of object or scene bias, as the same set of simple objects are used to render the videos. Fig. 2 describes the dataset and the associated tasks. We provide sample videos from the dataset in the supplementary video.

**Objects:** The CATER universe is built upon CLEVR (Johnson et al., 2017), inheriting most of the standard object shapes, sizes, colors and materials present in it. This includes three object shapes (cube, sphere, cylinder), in three sizes (small, medium, large), two materials (shiny metal and matte rubber) and eight colors, as well as a large "table" plane on which all objects are placed. In addition to these objects, we add two new object shapes: inverted cones and a special object called a 'snitch'. Cones also come in the same set of sizes, materials and colors. The 'snitch' is a special object shaped like three intertwined toruses in metallic gold color.

Figure 3: **Allen's temporal algebra.** Exhaustive list of temporal relations between intervals, as defined by Allen's algebra (Allen, 1983). For simplicity, we group them into three broad relations to define classes for composite actions, although in principle we could use all thirteen. Figure courtesy of (Alspaugh).

**Actions:** We define four atomic actions: 'rotate', 'pick-place', 'slide' and 'contain'; a subset of which is afforded by each object. The 'rotate' action means that the object rotates by 90°about its Y (or horizontal) axis, and is afforded by cubes, cylinders and the snitch. The 'pick-place' action means the object is picked up into the air along the Y axis, moved to a new position, and placed down. This is afforded by all objects. The 'slide' action means the object is moved to a new location by sliding along the bottom surface, and is also afforded by all objects. Finally, 'contain' is a special operation, only afforded by the cones, in which a cone is pick-placed on top of another object, which may be a sphere, a snitch or even a smaller cone. This allows for recursive containment, as a cone can contain a smaller cone that contains another object. Once a cone 'contains' an object, it is constrained to only 'slide' actions and effectively slides all objects contained within the cone. This holds until the top-most cone is pick-placed to another location, effectively ending the containment for that top-most cone.

**Animation process:** We start with an initial setup similar to CLEVR. A random number ($N$) of objects with random parameters are spawned at random locations at the beginning of the video. They exist on a $6 \times 6$ portion of a 2D plane with the global origin in the center. In addition to the random objects, we ensure that every video has a snitch and a cone. For the purposes of this work, we render 300-frame 320x240px videos, at 24 FPS, making it comparable to standard benchmarks (Soomro et al., 2012; Kuehne et al., 2011; Kay et al., 2017). We split the video into 30-frame slots, and each action is contained within these slots. At the beginning of each slot, we iterate through up to $K$ objects in a random order and attempt to add an action afforded by that object one by one without colliding with another object. As we describe later, we use $K = 2$ for our initial tasks and $K = N$ for the final task. For each action, we pick a random start and end time from within the 30-frame slot.

To further add to the diagnostic ability of this dataset, we render an additional set of videos with camera motion, with all other aspects of the data similarly distributed as the static camera case. For this, the camera is always kept pointed towards the global origin, and moved randomly between a predefined set of 3D coordinates. These coordinates include $X$ and $Y \in \{-10, 10\}$ and $Z \in \{8, 10, 12\}$. Every 30 frames, we randomly pick a new location from the Cartesian product of $X, Y, Z$, and move the camera to that location over the next 30 frames. However, we do constrain the camera to not change both $X$ and $Y$ coordinates at the same time, as that causes a jarring viewpoint shift as the camera passes over the $(0, 0, Z)$ point. Also, we ensure all the camera motion videos start from the same viewpoint, to make it easy to register the axes locations for localization task.

**Spatiotemporal compositions:** We wish to label our animations with the atomic actions present, as well as their compositions. Atomic actions have a well-defined spatiotemporal footprint, and so we can define composites using spatial relations ("a cylinder is rotating behind a sliding red ball"), similar to CLEVR. Unique to CATER is the ability to designate temporal relationships ("a cylinder rotates before a ball is picked-and-placed"). Because atomic actions occupy a well-defined temporal extent, we need temporal logic that reasons about relations between *intervals* rather than instantaneous events. While the latter can be dealt with timestamps, the former can be described with Allen's interval algebra with thirteen basic relations (Figure 3) along with composition operations. For simplicity, we group those into three broad relations. However, our dataset contains examples of all such interval relations and can be used to explore fine-grained temporal relationships.

## 3.1 TASKS DEFINED ON THE DATASET

Given this CATER universe with videos, ground truth objects and their actions at any time point, we can define arbitrarily complex tasks for a video understanding system. Our choice of tasks is informed by two of the main goals of video understanding: 1) Recognizing the *states of the actor*, including spatiotemporal compositions of those atomic actions. For example, a spatiotemporal composition of atomic human body movements can be described as an exercise or dance routine. And 2) Recognizing the effect of those actions on the *state of the world*. For example, an action involving picking and placing a cup would change the position of the cup and any constituent objects contained within it, and understanding this change in the world state would implicitly require understanding the action itself.

Given these two goals, we define three tasks on CATER. Each has progressively higher complexity, and tests for a higher level reasoning ability. To be consistent with existing popular benchmarks (Soomro et al., 2012; Kuehne et al., 2011; Kay et al., 2017; Sigurdsson et al., 2016), we stick to standard single or multi-label classification setup, with standard evaluation metrics, as described next. For each of these tasks, we start by rendering 5500 total videos, to be comparable in size with existing popular benchmarks (Kuehne et al., 2011). Since tasks 1 and 2 (defined next) explicitly require recognizing individual actions, we use $K = 2$ for the videos rendered to keep the number of actions happening in any given video small. For task 3, we set $K = N$ as the task is to recognize the end effect of actions, and not necessarily the actions themselves. We split the data randomly in 70:30 ratio into a training and test set. We similarly render a same size dataset with camera motion, and define tasks and splits in the same way as for the static camera. With the code release we also provide a further split of train set into a validation set (80:20). While we focus on the following tasks in this paper, note that the data is amenable to many other tasks, for instance (Malinowski et al., 2020) uses CATER for video reconstruction.

**Task 1: Atomic action recognition.** This first task on CATER is primarily designed as a simple debugging task, which should be easy for contemporary models to solve. Given the combinations of object shapes and actions afforded by them, we define 14 classes such as 'slide(cone)', 'rotate(cube)' and so on. Since each video can have multiple actions, we define it as a multi-label classification problem. The task is to produce 14 probability values, denoting the likelihood of that action happening in the video. The performance is evaluated using average precision per-class. Final dataset-level performance is computed by mean over all classes, to get mean average precision (mAP). This is a popular metric used in other multi-label action classification datasets (Sigurdsson et al., 2016; Gu et al., 2018).

**Task 2: Compositional action recognition.** While recognizing individual objects and motions is important, it is clearly not enough. Real world actions tend to be composite in nature, and humans have no difficulty recognizing them in whole or in parts. To that end, we construct a compositional action recognition task through spatiotemporal composition of the basic actions used in Task 1. For simplicity, we limit composites to pairs of 14 atomic actions, where the temporal relation is grouped into broad categories of 'before', 'during' and 'after' as shown in Figure 3. Combining all possible atomic actions with the three possible relations, we get a total of $14 \times 14 \times 3 = 588$ classes, and removing duplicates (such as 'X after Y' is a duplicate of 'Y before X'), leaves 301 classes. Similar to task 1, multiple compositions can be active in any given video, so we set it up as a multi-label classification problem, evaluated using mAP. If certain compositions never occur in the dataset, those are ignored for the final evaluation.

**Task 3: Snitch localization.** The final, and the flagship task in CATER, tests models' ability to recognize the effect of actions on the environment. Just as in the case of cup-and-ball trick, the ability of a model to recognize location of objects after some activity can be thought of as an implicit evaluation of its ability to understand the activity itself. The task now is to predict the location of the special object introduced above, the Snitch. While it may seem trivial to localize it from the last frame, it may not always be possible to do that due to occlusions and recursive containments. The snitch can be contained by other objects (cones), which can further be contained by other larger cones. All objects move together until 'uncontained', so the final location of the snitch would require long range reasoning about these interactions. For simplicity, we pose this as a classification problem by quantizing the $6 \times 6$ grid into 36 cells and asking which cell the snitch is in, at the end of the video. We ablate the grid size in experiments. Since the snitch can only be at a single location at the end of the video, we setup the problem as a single label classification, and evaluate it using standard

percentage accuracy metrics such as top-1 and top-5 accuracy. However, one issue with this metric is that is would penalize predictions where the snitch is slightly over the cell boundaries. While the top-5 metric is somewhat robust to this issue, we also report mean $L_1$ distance of predicted grid cell from the ground truth, as a metric that is congnizant of the grid structure in this task. Hence, it would penalize confusion between adjacent cells less than those between distant cells. The data is also amenable to a purely regression-style evaluation, though we leave that to future work.

## 4 EXPERIMENTS

We now experiment with CATER using recently introduced state of the art video understanding and temporal reasoning models (Carreira & Zisserman, 2017; Wang et al., 2018; 2016b; Hochreiter & Schmidhuber, 1997). I3D (Carreira & Zisserman, 2017), called R3D when implemented using a ResNet (He et al., 2016a) in (Wang et al., 2018), brings the best of image models to video domain by inflating it into 3D for spatiotemporal feature learning. Non-local networks (Wang et al., 2018) further build upon that to add a spatiotemporal interaction layer that gives strong improvements and out-performs many multi-stream architectures (that use audio, flow etc) on Kinetics and Charades benchmarks. For our main task, snitch localization, we also experiment with a 2D-conv based approach, Temporal Segment Networks (TSN) (Wang et al., 2016b), which another top performing method on standard benchmarks (Kay et al., 2017). This approaches uses both RGB and flow modalities. All these architectures learn a model for individual frames or short clips, and at test time aggregate the predictions by averaging over those clips.

While simple averaging works well enough on most recent datasets (Kay et al., 2017; Soomro et al., 2012; Kuehne et al., 2011), it clearly loses all temporal information and may not be well suited to our set of tasks. Hence, we also experiment with a learned aggregation strategy: specifically using an LSTM (Hochreiter & Schmidhuber, 1997) for aggregation, which is the tool of choice for temporal modelling in various domains including language and audio. We use a common LSTM implementation for aggregating either (Wang et al., 2016b) or (Wang et al., 2018) that operates on the last layer features (before logits). We extract these features for subclips from train and test videos, and train a 2-layer LSTM with 512 hidden units in each layer on the train subclips. The LSTM produces an output at each clip it sees, and we enforce a classification loss at the end, once the model has seen all the clips. At test time we take the prediction from the last clip as the aggregated prediction. We report the LSTM performance averaged over three runs to control for random variation. It is worth noting that LSTMs have been previously used for action recognition in videos (Donahue et al., 2015; Karpathy et al., 2014), however with only marginal success over simple average pooling. As we show later, LSTMs actually perform significantly better on CATER, indicating the importance of temporal reasoning.

For task 3, we also experiment with a state-of-the-art visual tracking method (Zhu et al., 2018). We start by using the GT information of the starting position of snitch, and project it to screen coordinates using the render camera parameters. We defined a fixed size box around it to initialize the tracker, and run it until the end of the video. At the last frame, we project the center point of the tracked box to the 3D plane (and eventually, the class label) by using a homography transformation between the image and the 3D plane. This provides a more traditional, symbolic reasoning baseline for our dataset, and as we show in results, is also not enough to solve the task. Finally, we do note that many other video models have been proposed in literature involving 2.5D convolutions (Tran et al., 2018; Xie et al., 2017), VLAD-style aggregation (Girdhar et al., 2017; Miech et al., 2017) and other multi-modal architectures (Wang et al., 2016a; Bian et al., 2017). We focus on the most popular and best performing models, and leave a more comprehensive study to future work. A random baseline is also provided for all tasks, computed as the average performance of random scores passed into the evaluation functions. Implementation details for all baselines are provided in the supplementary and code will be released.

**Task 1: Atomic action recognition:** Table 2 (a) shows the performance of R3D with and without the non-local (NL) blocks, using different number of frames in the clips. We use a fixed sampling rate of 8, but experiment with different clip sizes. Adding more frames helps significantly in this case. Given the ease of the task, R3D obtains fairly strong performance for static camera, but not so much for moving camera, suggesting potential future work in building models agnostic to camera motion.

Table 2: Performance on the (a) 14-way atomic actions recognition, (b) 301-way compositional action recognition, and (c) 36-way localization task, for different methods.

(a) Task 1 (Atomic)

| Camera | Model | NL | #frames | mAP |
|---|---|---|---|---|
| - | Random | - | - | 56.2 |
| Static | R3D | | 8 | 89.0 |
| Static | R3D | ✓ | 8 | 88.8 |
| Static | R3D | | 32 | 98.8 |
| Static | R3D | ✓ | 32 | 98.9 |
| Moving | R3D | | 8 | 82.4 |
| Moving | R3D | ✓ | 8 | 82.7 |
| Moving | R3D | | 32 | 90.5 |
| Moving | R3D | ✓ | 32 | 90.2 |

(b) Task 2 (Compositional)

| Camera | Model | NL | #frames | mAP Avg | mAP LSTM |
|---|---|---|---|---|---|
| - | Random | - | - | 19.5 | 19.5 |
| Static | R3D | | 8 | 39.5 | 52.1 |
| Static | R3D | | 32 | 44.2 | 53.4 |
| Static | R3D | ✓ | 32 | 45.9 | 53.1 |
| Static | R3D | | 64 | 43.7 | 43.5 |
| Moving | R3D | | 32 | 40.9 | 43.2 |
| Moving | R3D | ✓ | 32 | 41.1 | 43.5 |

(c) Task 3 (Localization)

| Camera | Model | #frames | SR | Avg Top 1 | Avg Top 5 | Avg $L_1$ | LSTM Top 1 | LSTM Top 5 | LSTM $L_1$ |
|---|---|---|---|---|---|---|---|---|---|
| - | Random | - | - | 2.8 | 13.8 | 3.9 | 2.8 | 13.8 | 3.9 |
| Static | Tracking | - | - | 33.9 | - | 2.4 | 33.9 | - | 2.4 |
| Static | TSN (RGB) | 1 | - | 7.4 | 27.0 | 3.9 | 15.3 | 50.0 | 3.0 |
| Static | TSN (RGB) | 3 | - | 14.1 | 38.5 | 3.2 | 25.6 | 67.2 | 2.6 |
| Static | TSN (Flow) | 1 | - | 6.2 | 21.7 | 4.4 | 7.3 | 26.9 | 4.1 |
| Static | TSN (Flow) | 3 | - | 9.6 | 32.2 | 3.7 | 14.0 | 43.5 | 3.2 |
| Static | R3D | 8 | 8 | 24.0 | 54.8 | 2.7 | 34.2 | 64.6 | 1.8 |
| Static | R3D | 16 | 8 | 26.2 | 56.3 | 2.6 | 24.2 | 48.9 | 2.5 |
| Static | R3D | 32 | 8 | 28.8 | 68.7 | 2.6 | 45.5 | 67.7 | 1.6 |
| Static | R3D | 64 | 8 | 57.4 | 78.4 | 1.4 | 60.2 | 81.8 | 1.2 |
| Static | R3D + NL | 32 | 8 | 26.7 | 68.9 | 2.6 | 46.2 | 69.9 | 1.5 |
| Moving | R3D | 32 | 8 | 23.4 | 61.1 | 2.5 | 28.6 | 63.3 | 1.7 |
| Moving | R3D + NL | 32 | 8 | 27.5 | 68.8 | 2.4 | 38.6 | 70.2 | 1.5 |

| Models | Kinetics | UCF-101 | HMDB-51 | CATER |
|---|---|---|---|---|
| 1 frame (RGB) (Donahue et al., 2015) | - | 67.4 | - | 7.4 |
| LSTM (RGB) (Donahue et al., 2015) | - | 68.2 | - | 15.3 |
| TSN (RGB) (Wang et al., 2016b) | 72.5 | 93.2 | 51.0 | 14.1 |
| TSN (Flow) (Wang et al., 2016b) | 62.8 | 95.3 | 64.2 | 9.6 |
| 2S I3D (Carreira & Zisserman, 2017) | 75.7 | 98.0 | 80.7 | - |
| 2S R(2+1)D (Tran et al., 2018) | 75.4 | 97.3 | 78.7 | - |
| R3D(+NL) (Wang et al., 2018) | 77.7 | - | - | 57.4 |

Table 3: **Long term reasoning.** Comparing the best reported performance of standard models on existing datasets and CATER (task 3). Unlike previous benchmarks, (1) temporal modeling using LSTM helps and (2) local temporal cues (flow) are not effective by itself on CATER. 2S here refers to 'Two Stream'. TSN performance from (Xiong, 2017; 2016).

**Task 2: Compositional action recognition:** Next we experiment with the compositional action recognition task. The training and testing is done in the same way as Task 1, except this predicts confidence over 301 classes. As evident from Table 2 (b), this task is harder for the existing models, presumably as recognizing objects and simple motions would no longer solve it, and models need to reason about spatiotemporal compositions as well. It is interesting to note that non-local blocks now add to the final performance, which was not the case for Task 1, suggesting modeling spatio-temporal relations is more useful for this task. LSTM aggregation also helps quite a bit as the model can learn to reason about long-range temporal compositions. As expected, moving camera makes the problem harder.

**Task 3: Snitch localization:** Finally we turn to the localization task. Since this is setup as a single label classification, we use softmax cross entropy loss to train and classification accuracy for evaluation. For tracking, no training is required as we use the pre-trained model from (Zhu et al., 2018) and run it on the validation videos. Table 2 (c) shows the performance of various methods, evaluated at different clip lengths and frame rates. For this task we also experiment with TSN (Wang et al., 2016b), though it ends up performing significantly worse than R3D. Note that this contrasts with standard video datasets (Kay et al., 2017), where it tends to perform similar to R3D (Xiong, 2017). We also experiment with the flow modality and observe it obtains even lower performance, which is expected as this task requires recognizing objects which is much harder from flow. Again, note that flow models obtain similar if not better performance as RGB on standard datasets (Kay et al., 2017; Xiong, 2017). We also note higher performance on considering longer clips with higher sample rate. This is not surprising as a task like this would require long term temporal reasoning, which is aided by looking at longer videos. This is also reinforced by the observation that using LSTM for aggregation leads to a major improvement in performance for most models. Finally, the tracking approach also only solves about a third of the videos, as even the state of the art tracker ends up drifting due to occlusions and contain operations.

In Table 4, we ablate the performance with respect to the underlying grid granularity, with $6 \times 6$ being the default used in Table 2 (c). We observe tracking is a stronger baseline as the localization task gets more fine-grained. Finally in Table 3 we compare performance of some of these models on existing benchmarks and CATER.

Table 4: **Task 3 grid resolution:** Top-1 accuracy of our main baselines on changing the grid resolution. As expected, the overall performance improves when considering a coarser grid, while tracking becomes a stronger baseline for fine-scale localization.

| Model | $4 \times 4$ | $6 \times 6$ | $8 \times 8$ |
|---|---|---|---|
| R3D+NL (Avg) | 34.5 | 26.7 | 20.5 |
| R3D+NL (LSTM) | 56.4 | 46.2 | 17.5 |
| Tracking | 41.8 | 33.9 | 28.6 |

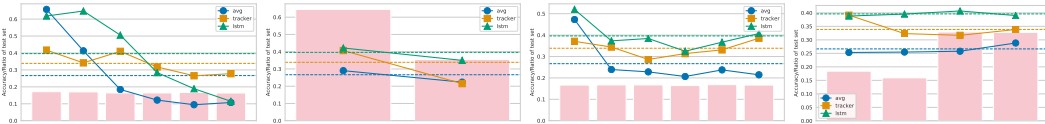

(a) When snitch last moved (b) Snitch contained at end (c) Displacement of snitch (d) Number of objects

Figure 4: **Diagnostic analysis of localization performance.** We bin the test set using certain parameters. For each, we show the test set distribution with the bar graph, the performance over that bin using the line plot, and performance of that model on the full val set with the dotted line. We find that localization performance, (a) Drops significantly if the snitch is kept moving till the end. This is possibly because for cases when snitch only moves in the beginning and is static after, the models have a lot more evidence to predict the correct location from. Interestingly the tracker is much less affected by this, as it tracks the snitch until the very end; (b) Drops if the snitch is 'contain'-ed by another object in the end, and the tracker is the worst affected by it; (c) Drops initially with increasing displacement of the snitch from its start position, but is stable after that; and (d) Is relatively stable with different number of objects in the scene.

**Analysis:** Having close control over the dataset generation process enables us to perform diagnostics impossible with any previous dataset. We use the R3D+NL, 32-frame, static camera model with average (or LSTM, when specified) pooling for all following visualizations. We first analyze aggregate performance of our model over multiple bins in Figure 4, and observe some interesting phenomena. *(a) Performance drops if the snitch keeps moving until the end.* This makes sense: if the snitch reaches its final position early in the video, models have a lot more frames to reinforce their hypothesis of its final location. Between LSTM and avg-pooling, LSTM is much better able to handle the motion of the snitch, as expected. Perhaps not surprisingly, the tracker is much less effected by snitch movement, indicating the power of such classic computational pipelines for long-term spatiotemporal understanding. *(b) Drops if the snitch is contained in the end.* Being contained in the final frame makes the snitch harder to spot and track (just like the cups and ball game!), hence the lower performance.

Next, we visualize the videos that our models gets right or wrong. We sort all validation videos based on the softmax confidence score for the ground truth class, and visualize the top and bottom six in Figure 5 (full video in **supplementary**). We find that the easiest videos for avg-pooled model tend to be ones with little snitch motion, i.e. the object stays at the position it starts off in. On the other hand, the LSTM-aggregated model fares better with snitch motion, as long as it happens early in the video. The hardest videos for both tend to be ones with sudden motion of the snitch towards the end of the video, as shown by the bright golden trail denoting the motion towards the end (better viewed in supplementary video). These observations are supported by the quantitative plots in Figure 4 (a) and (c).

## 5 CONCLUSION

We use CATER to analyze several leading network designs on hard spatiotemporal tasks. We find most models struggle on our proposed dataset, especially on the snitch localization task which requires long term reasoning. Interestingly, average pooling clip predictions or short temporal cues (optical flow) perform rather poorly on CATER, unlike most previous benchmarks. Such temporal reasoning challenges are common in the real world (eg. Fig. 1 (a)), and solving those would be the cornerstone of the next improvements in machine video understanding. We believe CATER would

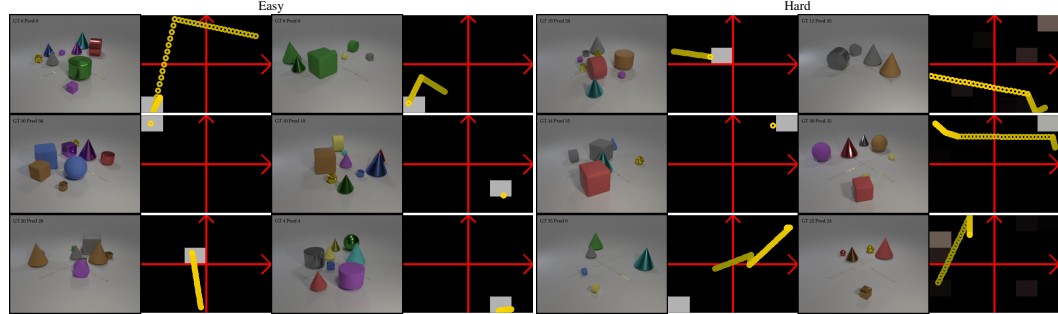

Figure 5: **Easiest/hardest videos for localization.** We analyze the top most confident a) correct and b) incorrect predictions on the test videos for localization task. For each video, we show the last frame, followed by a top-down view of the $6 \times 6$ grid. The grid is further overlaid with: 1) the ground truth positions of the snitch over time, shown as the golden trail, which fades in color over time $\implies$ brighter yellow depicts later positions; and 2) the softmax prediction confidence scores for each location (black is low, white is high). The model has easiest time classifying the location when the snitch does not move much or moves early on in the video. Full video in **supplementary**.

serve as an intermediary in building systems that will reason over space and time to understand actions. That said, CATER is, by no means, a complete solution to the video understanding problem. Like any other synthetic or simulated dataset, it should be considered in addition to real world benchmarks. While we have focused on classification tasks for simplicity, our fully-annotated dataset can be used for much richer parsing tasks such as spacetime action localization. One of our findings is that while high-level semantic tasks such as activity recognition may be addressable with current architectures given a richly labeled dataset, "mid-level" tasks such as tracking still pose tremendous challenges, particularly under long-term occlusions and containment. We believe addressing such challenges will enable broader temporal reasoning tasks that capture intentions, goals, and causal behavior.

### ACKNOWLEDGMENTS

Authors would like to thank Ishan Misra for many helpful discussions and help with systems. This research is based upon work supported in part by NSF Grant 1618903.

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

## A    IMPLEMENTATION DETAILS FOR BASELINES

We use the provided implementation for ResNet-3D (R3D) and non-local (NL) block from (Wang et al., 2018), and temporal segment networks (TSN) from (Wang et al., 2016b) for all our experiments. For (Wang et al., 2018), all the models are based on ResNet-50 base architecture, and trained with hyperparameters scaled down from Kinetics as per CATER size. For non-local (NL) experiments, we replace the conv3 and conv4 blocks in ResNet with the NL blocks. All models are trained with classification loss implemented using sigmoid cross-entropy for Task 1 and 2 (multi-label classification task), and softmax cross-entropy for task 3. At test time, we split the video into 10 temporal clips and 3 spatial clips. When aggregating using average pooling, we average the predictions from all 30-clips. For LSTM, we train and test on the 10 center clips. We experiment with varying the number of frames (#frames) and sampling rate (SR). For TSN (Wang et al., 2016b), the model is based on BN-inception (Szegedy et al., 2016), with hyperparamters following their implementation on HMDB (Kuehne et al., 2011) given its similar size and setup to our dataset. For optical flow we use the TVL1 (Sánchez Pérez et al., 2013) implementation. At test time we aggregate the predictions over 250 frames uniformly sampled from the video, either by averaging or using LSTM. While CATER videos look different from real world, we found the networks much easier to optimize with the ImageNet initialization for both approaches. This is consistent with prior work (Wang et al., 2016b) that finds ImageNet initialization is useful even when training diverse modalities (such as optical flow). We will make the code, generated data and models available for more implementation details.

## B    TRAIN/VAL DISTRIBUTIONS

Figure 6 shows the data distribution over classes for each of the tasks.

## C    VIDEO VISUALIZATION

The supplementary video[3] visualizes:

1. **Sample videos** from the dataset (with and without camera motion).
2. **Easiest and hardest videos for task 3.** We rank all validation videos for task 3 based on their softmax probability for the correct class. We show the top-6 (easiest) and bottom-6 (hardest) for 32-frame stride-8 non-local + LSTM model. We observe the hardest ones involve sudden motion towards the end of the video. This reinforces the observation made in Figure 5(a) in the main paper, that videos where snitch keeps moving till the end are the hardest. If the snitch stops moving earlier, models have more evidence for the final location of the snitch, making the task easier.
3. **Tracking results.** We visualize the results of tracking the snitch over the video as one approach to solving task 3. We observe that while it works in the simple scenarios, it fails when there is a lot of occlusion or complex contain operations.
4. **Model bottom-up attention.** We visualize where does the model look for Task 3. As suggested in (Malinowski et al., 2018), we visualize the $l_2$-norm of the last layer features from our 32-frame stride-8 non-local model on the center video crop. The deep red color denotes large norm value at that spatiotemporal location. We find that the model automatically learns to focus on the snitch towards the end of clips, which makes sense as that is the most important object for solving the localization task.

---

[3] https://rohitgirdhar.github.io/CATER/assets/suppl/video.mp4

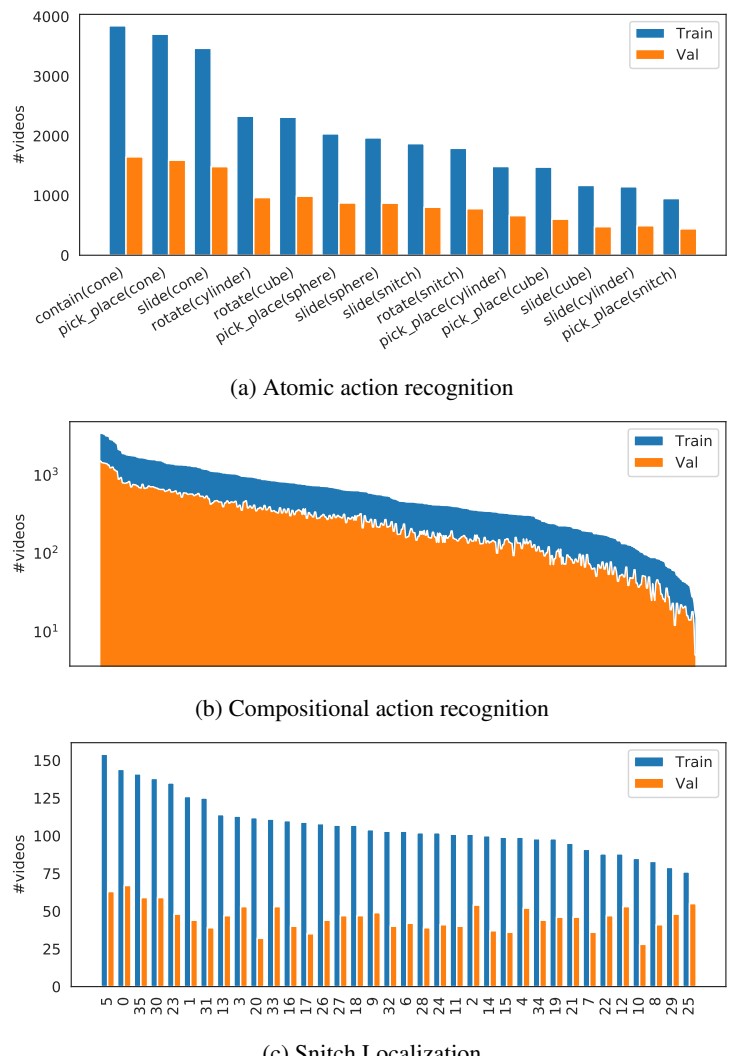

(a) Atomic action recognition

(b) Compositional action recognition

(c) Snitch Localization

Figure 6: **Train/val distribution.** Histograms of training and validation data distribution for different tasks we define on the dataset. (a) requires the model to recognize atomic actions, such as 'a sphere slides'. We defined 13 such classes. (b) requires recognizing spatiotemporal compositions of actions, such as 'a sphere slides while a cube rotates'. Since there are a total of 588 combinations, we omit the labels here for ease of visualization. Finally (c) evaluates the snitch localization task, where the model needs to answer where the snitch is on the board, quantized into a $6 \times 6$ grid, at the end of the video. This is defined as a 36-way classification problem.

