# OpenReview forum: "CATER: A diagnostic dataset for Compositional Actions & TEmporal Reasoning"
_ICLR.cc/2020/Conference — Accept (Talk)_

### Official Review · AnonReviewer3 · 2019-10-23
**Official Blind Review #3**

**Rating:** 8

**Review:**

This paper introduces a new synthetic video understanding dataset, borrowing many ideas from the visual question answering dataset CLEVR. The new dataset is the first to account for all of the following fundamental aspect of videos: temporal ordering, short- and long term reasoning, and control for scene biases. Due to the inherent biases in available action recognition datasets, models that simply averages video frames do nearly as well as models that take temporal dependencies into account. In contrast, the authors show that with the proposed dataset, models without spatiotemporal reasoning largely fail.

The paper should be accepted as it addresses a major shortcoming of all existing video understanding datasets. It does a good job at summarizing the deficiencies in existing datasets, clearly motivating the need for a new dataset. The claims are backed up with solid experiments, ablating models and data parameters adequately. It is mostly well-written (except for section 4 which would benefit from extensive proofreading) and does a good job at covering relevant work. One drawback is of course the synthetic nature and limited domain of objects and actions. On the other hand, this makes the setup highly controllable and reliable. I like the fact that each task comes both with both static and moving camera.

Improvements and Questions:
Some relevant datasets are missing. For example, the Moving MNIST and Robot Pushing datasets could be added to Table 1.

I suggest having a train / validation / test split (like CLEVR), rather than just a train and validation split.

In particular for Task 3 more frames seem to give dramatic improvement. Why did you not run with more than 64 frames?

Did you consider downsampling the videos to allow running on all the frames?

I’m missing details on the resolution of the generated videos?


**Experience Assessment:**

I have read many papers in this area.

**Review Assessment: Checking Correctness Of Derivations And Theory:**

N/A

**Review Assessment: Checking Correctness Of Experiments:**

I assessed the sensibility of the experiments.

**Review Assessment: Thoroughness In Paper Reading:**

I read the paper at least twice and used my best judgement in assessing the paper.

---

> ### Author Response · Authors · 2019-11-11
> **Response to Reviewer 3**
>
> Thank you for your time and insightful feedback! We have incorporated the changes into the revised paper, and address the issues in detail here:
>
> - Additional dataset comparisons: Thanks for pointing those out! We have added additional datasets to Table 1.
>
> - Train-Val-Test set: We will add that to the code release and update the final paper accordingly.
>
> - Running with more frames, downsampling videos, resolution: That’s a great point! CATER videos are rendered at 320x240px (updated in the paper) to be comparable to existing benchmarks, so standard baselines can be applied directly without significant redesign. 64 frames were the maximum we were able to fit in the GPU memory (at the extreme batch size of 1 per 12GB GPU), so scaling the number of frames beyond that was not possible given our current GPU resources. While reducing the frame size to fit all 300 frames might be possible, it would require redesigning and tuning the network architectures for the baselines since many hyper-parameters (kernel sizes, number of layers etc) would change for the super low resolution videos. Perhaps more importantly,  we believe our analysis suggests that in order to enable truly long-term temporal reasoning, one should rely on representations that more efficiently maintain stateful memory, such as recurrent networks / LSTMs.

---

### Official Review · AnonReviewer2 · 2019-10-27
**Official Blind Review #2**

**Rating:** 8

**Review:**

The paper introduces CATER: a synthetically generated dataset for video understanding tasks. The dataset is an extension of CLEVR using simple motions of primitive 3D objects to produce videos of primitive actions (e.g. pick and place a cube), compositional actions (e.g. "cone is rotated during the sliding of the sphere"), and finally a 3D object localization tasks (i.e. where is the "snitch" object at the end of the video).  The construction of the dataset focuses on demonstrating that compositional action classification and long-term temporal reasoning for action understanding and localization in videos are largely unsolved problems, and that frame aggregation-based methods on real video data in prior work datasets, have found relative success not because the tasks are easy but because of dataset bias issues.

A variety of models from recent work are evaluated on the three proposed tasks, demonstrating the validity of the above motivation for the construction of the dataset.  The primitive action classification task is "solved" by nearly all methods and only serves for debugging purposes.  The compositional action classification task is harder and shows that incorporating LSTMs for temporal reasoning leads to non-trivial performance improvements over frame averaging.  Finally, the localization task is challenging, especially when camera motion is introduced, with much space for improvement left for future work.

I am positive with respect to acceptance of this paper.  It is a well-argued, thoughtful dataset contribution that sets up a reasonable video understanding dataset.  The authors recognize that since the dataset is synthetically generated it is not necessarily predictive of how methods would perform with real-world data, but still it can serve a useful and complementary role similar to the one CLEVR has served in image understanding.

I have a few minor comments / questions / editing notes that would be good to address:
- The random baseline isn't described in the main text, it would be good to briefly mention it (this will also help to clarify why the value is particularly high for tasks 1 and 2)
- The grid resolution ablation results presented in the supplement are actually quite important -- they demonstrate that with a small increase in granularity of the grid the traditional tracking methods begin to be the best performers. As this direction (of increased resolution to make the problem less artificial) is likely to be important, a brief discussion of this finding from the main paper text would be appropriate
- p3 resiliance -> resilience
- p4 objects is moved -> object is moved
- p6 actions itself -> actions themselves; builds upon -> build upon
- p7 looses all -> loses all; suited our -> suited to our; render's camera parameters -> render camera parameters; to solve it -> to solve the problem
- p8 (Xiong, b;a) and (Xiong, b) -> these references are missing the year; models needs to -> models need to
- p9 phenomenon -> phenomena; the the videos -> the videos; these observation -> these observations; of next -> of the next; in real world -> in the real world


**Experience Assessment:**

I have published one or two papers in this area.

**Review Assessment: Checking Correctness Of Derivations And Theory:**

I assessed the sensibility of the derivations and theory.

**Review Assessment: Checking Correctness Of Experiments:**

I assessed the sensibility of the experiments.

**Review Assessment: Thoroughness In Paper Reading:**

I read the paper at least twice and used my best judgement in assessing the paper.

---

> ### Author Response · Authors · 2019-11-11
> **Response to Reviewer 2**
>
> Thank you for your time and insightful feedback! We have incorporated the changes into the revised paper, and address the issues in detail here:
>
> - Random baseline: We compute those by providing random scores to the evaluation function and averaging the performance output over multiple runs. This ensures that the random performance takes into account any biases in the label distribution. For task 1 and 2, the evaluation is performed using mAP (due to the multi-label setup), which essentially evaluates a ranking problem and hence is a more forgiving metric than classification. For example, ranking the correct prediction at 2nd position would still get a good mAP, but would get a 0% classification accuracy.
>
> - Grid resolution: That’s a great point, we’ve moved the discussion to the main paper.
>
> - Text issues: Thanks a lot for pointing those out! We have fixed the revised version.

---

### Official Review · AnonReviewer4 · 2019-11-03
**Official Blind Review #4**

**Rating:** 8

**Review:**

This paper proposed a new synthetic dataset (CATER) for video understanding. The authors argue that since current video datasets are heavily biased over static scenes and object structures, it is unclear whether modern spatial-temporal video models can learn to reason over temporal dimension. In order to address this problem, they design this fully observable synthetic dataset which is built upon CLEVER, along with three tasks that are customized for temporal understanding. They further conduct a variety of experiments to benchmark state-of-the-art video understanding models and show how those models more or less struggle on temporal reasoning.

Overall this paper is well-written and easy to follow. The problem is well-motivated, and the claims are mostly supported. The diagnosis in this paper provides useful insights that could be contributive to both vision and learning communities.

My primary concern is to what extent can the new dataset (CATER) add to existing video datasets that are also explicitly designed for long term spatial-temporal reasoning, such as video VQA datasets TGIF-QA[1]/SVQA[2]. In addition to the comparison between CATER and three action recognition datasets (Kinetics/UCF101/HMDB51) as presented in Table 3., it would be more interesting to see how video understanding models that are specifically designed for those video VQA datasets will perform on CATER.

[1] Yunseok Jang, Yale Song, Youngjae Yu, Youngjin Kim, and Gunhee Kim. Tgif-qa: Toward spatio-temporal reasoning in visual question answering. In Proceedings of the IEEE Conference on Computer Vision and Pattern Recognition, pages 2758–2766, 2017.
[2] Xiaomeng Song, Yucheng Shi, Xin Chen, and Yahong Han. Explore multi-step reasoning in video question answering. In 2018 ACM Multimedia Conference on Multimedia Conference, pages 239–247. ACM, 2018.

**Experience Assessment:**

I have published one or two papers in this area.

**Review Assessment: Checking Correctness Of Derivations And Theory:**

N/A

**Review Assessment: Checking Correctness Of Experiments:**

I carefully checked the experiments.

**Review Assessment: Thoroughness In Paper Reading:**

I read the paper at least twice and used my best judgement in assessing the paper.

---

> ### Author Response · Authors · 2019-11-11
> **Response to Reviewer 4**
>
> Thank you for your time and insightful feedback!
> That’s a great point regarding comparison with video-QA datasets. While the motivation for such datasets is also long-term spatiotemporal reasoning, it is typically hard to implement them in an unbiased manner. As a recent study from ICCV’19 showed (https://bhavanj.github.io/MovieQAWithoutMovies/), it is possible to outperform all previous video QA methods on the MovieQA benchmark without actually looking at the videos (simply by looking at the question and answer choices). One reason for this is that a language/QA interface adds additional complexity and opportunities for linguistic biases to creep into the task. While SVQA takes an important step towards controlling these biases through synthetic videos, our goal with CATER is to primarily focus on the video understanding problem, and hence we design it with a simple classification task. That being said, CATER provides the full ground truth state of the world at each time instant, and so can be used to generate QA pairs either automatically (akin to CLEVR or SVQA) or by humans. Such a QA interface will also enable testing of video-QA baselines as suggested, since those are typically designed for jointly modeling linguistic and visual elements. We leave a full exploration of such a QA interface to future work, however we have added these datasets and comparison to Table 1 in the revised paper.

---

### Public Comment · ~Yingwei_Li1 · 2020-02-01
**quite similar to our previous idea**

http://openaccess.thecvf.com/content_ECCV_2018/html/Yingwei_Li_RESOUND_Towards_Action_ECCV_2018_paper.html

---

> ### Author Response · Authors · 2020-02-02
> **re: quite similar to our previous idea**
>
> Thank you for the comment! We cite this paper in our final version. And yes, there have been some recent works pushing towards controlling biases in video datasets, though complementary to those CATER provides additional diagnostic ability and control (for eg, length of videos for long-range reasoning) which is hard to achieve with most previous benchmarks.

---

### Decision · Program_Chairs · 2019-12-19

**Decision:**

Accept (Talk)

**Comment:**

The paper proposed a new synthetically generated video dataset (CATER) for benchmarking temporal reasoning. The dataset is based on the CLEVR dataset and provides videos make up of primitive actions ("rotate", "pick-place", "slide", "contain") that can be combined to form for complex actions.
The paper also benchmarks a variety of methods on three proposed tasks (atomic action classification, composite action classification, and 'snitch' localization) and demonstrates that while it is possible to get high performance on atomic action classification, the other two task are still challenging and requires temporal modeling.

Overall, all reviewers found the paper to be well written and easy to follow, with care given to the dataset construction, as well as the task definitions and experiment setup and analysis.  The paper received strong scores from all reviewers (3 accepts).  Based on the reviewer comments, the authors further improved the paper by adding additional relevant datasets for comparison and providing missing details pointed out by the reviewers.  After the rebuttal, the reviewers remained positive.